# A Sensor-Based mHealth Platform for Remote Monitoring and Intervention of Frailty Patients at Home

**DOI:** 10.3390/ijerph182111730

**Published:** 2021-11-08

**Authors:** Jorge Calvillo-Arbizu, David Naranjo-Hernández, Gerardo Barbarov-Rostán, Alejandro Talaminos-Barroso, Laura M. Roa-Romero, Javier Reina-Tosina

**Affiliations:** 1Biomedical Engineering Group, University of Seville, 41092 Seville, Spain; dnaranjo@us.es (D.N.-H.); gbarbarov@gmail.com (G.B.-R.); talaminos@gmail.com (A.T.-B.); lroa@us.es (L.M.R.-R.); jreina@us.es (J.R.-T.); 2Department of Telematics Engineering, University of Seville, 41092 Seville, Spain; 3Department of Signal Theory and Communications, University of Seville, 41092 Seville, Spain

**Keywords:** mHealth, frailty, health information technology, sensors, fall detection, activity recognition

## Abstract

Frailty syndrome is an independent risk factor for serious health episodes, disability, hospitalization, falls, loss of mobility, and cardiovascular disease. Its high reversibility demands personalized interventions among which exercise programs are highly efficient to contribute to its delay. Information technology-based solutions to support frailty have been recently approached, but most of them are focused on assessment and not on intervention. This paper describes a sensor-based mHealth platform integrated in a service-based architecture inside the FRAIL project towards the remote monitoring and intervention of pre-frail and frail patients at home. The aim of this platform is constituting an efficient and scalable system for reducing both the impact of aging and the advance of frailty syndrome. Among the results of this work are: (1) the development of elderly-focused sensors and platform; (2) a technical validation process of the sensor devices and the mHealth platform with young adults; and (3) an assessment of usability and acceptability of the devices with a set of pre-frail and frail patients. After the promising results obtained, future steps of this work involve performing a clinical validation in order to quantify the impact of the platform on health outcomes of frail patients.

## 1. Introduction

According to the World Health Organization (WHO), the worldwide population of people over 60 years of age will nearly double between 2015 and 2050, from 12% to 22%, increasing from 900 million to 2 billion [1]. Besides chronicity and illnesses, aging leads to symptoms of physical and cognitive decline that have been grouped under the geriatric concept of frailty syndrome [2]. Approximately 11% of community dwelling adults aged 65 years and over suffers from frailty, and it is more common in elderly women [3]. Several definitions of this syndrome coexist, but all concur that frailty represents a continuum between a healthy elderly person and an extremely vulnerable one, with a high risk of death and low odds of recovery. In this continuum, frail and pre-frail adults can be identified. Whereas the former have a higher risk of disability, resistance loss, and vulnerability to adverse events leading to increased morbidity and mortality, pre-frail persons suffer mild symptoms, and oriented interventions may slow down the decline process [4,5].

Frailty is an independent risk factor for serious health episodes, a predictor of disability, hospitalization, falls, loss of mobility and cardiovascular disease. Its high reversibility demands personalized interventions depending on the scenario of each user (e.g., home or long-term care facilities) [6]. Among interventions, exercise programs are highly efficient at reducing fragility, since avoiding inactivity (both physically and cognitively or emotionally) may contribute to its prevention, delay, and even reversion.

Connected health and information technology (HIT) are disrupting the way healthcare is being delivered, and HIT-based solutions have been proposed in the last years to support frailty. Most approaches are focused on assessment, while intervention on frailty is barely reported [7] or is mainly focused on nutrition, cognitive, and physical training [8,9,10]. Wearable sensors at home or in hospital settings are often used to assess subject’s activity [11,12,13,14,15], but other approaches such as smart-cities are also applied for assessing early frailty symptoms [16]. Additionally, any HIT-solution devoted to elderly people should be designed specifically for this population, since it is recognized that technology anxiety hinders the acceptance of mHealth solutions and continuance use [17].

Despite the existing approaches, the absence of effective interventions on frailty syndrome encourages the research and development of connected health solutions that support care of frail citizens, reducing deterioration caused by aging and improving their quality of life. This paper presents a sensor-based mHealth platform developed inside the FRAIL project [18] that proposes a multidimensional intervention based on HIT for frail and pre-frail citizens by combining sensor devices with gamification tools as well as services devoted to formal and informal caregivers. The aim of this platform is constituting an efficient and scalable system for reducing both the impact of aging and the advance of frailty syndrome.

The rest of the paper is structured as follows. Section 2 describes frail population needs and their translation as design requirements of the mHealth platform. Section 3 presents the FRAIL project where the platform is integrated, the architecture and components of which is presented in Section 4. Section 5 describes results from usability testing, and Section 6 the conclusions of this work.

## 2. User Needs of Citizens in Risk of Frailty

Frail patients experience reduction of muscle mass and strength, what leads to physical insecurity with involuntary movement functions (e.g., impaired balance), and, consequently, decline of daily life activities. Thus, the most common issues related to frailty are mobility limitations and falls. Resistance and strength exercise interventions can improve muscle mass and reduce the risk of falling in elderly people living in both community and nursing care settings.

From the identified needs and recommendations provided by both the National Institute for Health and Care Excellence (NICE) [19] and the European Innovation Partnership on Active and Healthy Ageing (EIP-AHA) on frailty (Action Group 3) [20], a set of technology requirements can be established (Table 1). Sensor devices may address those needs, including monitoring heart and breathing rate, movements and physical exercise performance, and detection of falls. Monitored data and events should be available for authorized (formal and informal) caregivers, joining remote supervision with daily care. Finally, the potential delay (even reversibility) of frailty syndrome stimulates the deployment of exercise interventions where the subject is actively involved.

## 3. The FRAIL Project

The sensor-based platform presented here has been designed and developed inside the Eurostars-funded project FRAIL with the aim of designing and building an interoperable, secure, and open solution to empower pre-fragile and fragile citizens through a non-intrusive sensing of activity and environmental conditions in the home context [18]. The FRAIL architecture is a suite of three interconnected elements: (1) a personal and home sensing platform (the focus of this work), (2) an intervention system for care professionals, and (3) a gamification system for physical and mental exercise devoted to elderly people. Figure 1 shows the architecture of FRAIL. Frail patients may use FRAIL with or without the assistance of a caregiver, whereas formal or informal caregivers participate as delegates of patients.

The gateway of the sensor-based mHealth platform (i.e., the smartwatch) is responsible for sending/receiving data to/from the other FRAIL systems. Whereas the intervention system leaves it to health professionals to prescribe activity care plan to patients, the gamification system provides to frail users with interactive games and videos for encouraging the activities performance. Activity monitoring data are used to assess the level of adherence to the care plan and the adequate performance of exercises. Intervention and gamification systems are accessed via user web interfaces, separated for professionals and users, whereas the sensing platform is interactive, using sensors and smartwatch. In the following, the sensor-based mHealth platform and its devices are described in detail.

## 4. Personal and Home Sensing Platform

The sensing platform aims to address the continuous monitoring of vital signs relevant for pre-/frail users as well as detecting and alerting falls. A smartwatch worn by the frail user acts as a gateway to the platform, gathering data from sensors and receiving events and reminders introduced by caregivers in the intervention system (Figure 2). Since the smartwatch is responsible for the communication with the FRAIL servers, it eases the deployment of different sensor devices that only need to communicate with it. Initially, two devices were incorporated into the mHealth platform: a falls detector and smart vest for physical exercise and respiratory monitoring. Although the sensing platform is designed for scenarios at home, the connectivity of the smartwatch also allows for the monitoring of outdoor events. The main components of the sensing mHealth platform are described below.

### 4.1. Fall Detection

Early alerting of falls is a major requirement for elderly people living alone. Fall detection has been addressed through an accelerometry-based device that monitors a person’s movement and performs a first detection of impacts (may be falls or not) that are sent to the smartwatch. This assesses them in order to decide whether they correspond to a true fall or another kind of impact. At the same time, the smartwatch asks the subject whether she needs help. In any case, after a few seconds, if the user does not choose any option (because she is disabled) a warning will automatically be sent to the caregivers.

The distributed processing in two stages (first the device and then the smartwatch) reduces energy consumption of the portable device and extends battery life. If a fall is detected, the smartwatch sends the event to FRAIL servers and triggers pre-configured procedures. Thereby, the fall detection mechanism consists of a distributed processing architecture composed of two components: sensor (hardware) and API (software).

#### 4.1.1. Smart Sensor for the Impact Detection

This small device is composed in turn by:

The impact detection device itself, which is the persistent part made up of the hardware and a protective shell. The device hardware is supported by the following modular design (Figure 3):

Sensor module: the movement of the human body is monitored by a triaxial accelerometer (LIS3LV02DQ from STMicroelectronics).

Processing module: the operation of the device is managed by a microcontroller (PIC18F2431 from Microchip). In it, an algorithm for the detection of impacts is executed from the triaxial acceleration samples captured with a sampling frequency of 40 Hz. The characteristics of the impact detection algorithm were described in [21]. When an impact is detected, the acceleration samples corresponding to two seconds before and two seconds after the impact are sent wirelessly to a second device with higher processing capabilities. This significantly reduces the power consumption of the device since data are only sent when an impact is detected. The impacts may or may not correspond to a fall, and a more computationally complex algorithm executed on a higher-level device decides whether the impact corresponds to a true fall.

Communication module: Wireless communications are bidirectional and are based on the Bluetooth 4.1 standard over the GATT protocol. These communications have been implemented in an RN4020 transceiver from Microchip. A read-type characteristic is used to send the accelerations, which allows the transfer of data in blocks of 20 bytes. A write-enabled feature for the transfer of one-byte blocks is used for command reception.

An adhesive patch is placed on the skin of the lower back. This position and the design of the patch provide the fall detector with a set of advantages in terms of usability and acceptance. Typical fall detection devices on bracelets or pendants are more sensitive to motion artifacts, thus generating a high number of false positives. In addition, these devices are in visible positions, and users are reluctant to use them to avoid stigmatization that comes with being elderly. Our device is in a position close to the centre of gravity, providing robustness against motion artifacts, increasing sensitivity and specificity, and reducing the number of false positives. In addition, the results of surveys to the elderly reaffirm the adequacy of this position as it is in a non-visible location, which increases user acceptance since it does not interfere with social interactions. The patch is biocompatible and disposable, shaped in such a way that it allows the device to be attached inside (Figure 4). In addition, the battery that supplies the device is integrated inside it, so that the connector of the device with the battery serves as a fixing element between the device and the adhesive patch. The fixation element of the patch on the skin is a hypoallergenic adhesive made of hydrocolloid material. Considering that for hygiene reasons the patch must be replaced every seven days, the autonomy of the battery has been dimensioned for the device to operate between seven and 10 days. Finally, a transparent waterproofing layer of 10 cm × 10 cm of polyurethane suitable for application on the skin ensures the fixed position of the patch on the back and serves to completely isolate the device from water and moisture in activities such as taking a shower. A usability test carried out on a 38-year-old young man showed the suitability of the adhesive patch for four-days of continuous use. The volunteer reported being aware of having the patch in place during the first hours, but also indicated that after this period he forgot that he had it on his back, not interfering with daily activities such as sitting, lying in bed, or taking a shower. After the first hours of use, the volunteer confirmed the comfort of the proposed solution, which was maintained for the rest of the experiment. The impermeability, adherence and fixation of the patch were maintained throughout the usability test. A biocompatible adhesive patch is a novel solution to fix a fall detection device on the human body for as long as necessary, with the added value that it avoids the possibility of the person forgetting to put it on, a very common situation in fall detectors, protecting users even while they are taking a shower or in bed, where many falls occur.

#### 4.1.2. The Fall Detection Application Program Interface (API) in the Smartwatch

Wireless communications are managed with the impact detection device using the GAP protocols for establishing the connection and GATT for receiving accelerometry data. The API also executes the high-level algorithm for the detection of fall on the acceleration samples related to an impact. The characteristics of the fall detection algorithm were described in [22]. The API simplifies and abstracts from all the management functions of the fall detection system and provides the interface for the management of fall events to the control applications implemented in the smartwatch.

### 4.2. Smart Vest

Remote monitoring of physical exercise and respiratory parameters is performed by means of a wearable smart vest. This vest is an evolution of the design presented in [23], which incorporates physical activity monitoring and usability improvements for use by elderly people. Similar to the fall detector, the smart vest also has a distributed processing architecture composed of several components (Figure 5).

#### 4.2.1. Smart Device for Respiratory and Physical Activity Monitoring

This device integrates the hardware in a casing that can be easily removed from the garment to facilitate washing. The device hardware also follows a modular design architecture consisting of:

Respiratory sensorization module: Respiratory movements are captured through variations in the frequency of oscillation of a Colpitts oscillator, whose frequency of oscillation depends on the electrical capacity between two electrodes made of a conductive textile material (CN-3190 from 3M) with rectangular shape (25 cm × 13 cm), integrated inside the smart vest and located one on the back and one on the front of the chest. The electrodes are connected to the smart monitoring device using cables built into the inside of the smart vest and a connector that makes it easy to dock.

Physical activity sensorization module: The movements of the human body during elderly maintenance exercises are recorded using a triaxial accelerometer (LIS3LV02DQ from STMicroelectronics).

Processing module: Considering that both respiratory movements and accelerations during maintenance exercises have sawteeth-like waveforms, the signals captured by the respiratory sensing module and the physical activity sensing module are processed to abstract the relevant information from the waveform using the algorithm described in [24,25]. This reduces the amount of data sent wirelessly, thus reducing power consumption.

Interface module: The device includes a push button for on and off, a LED indicator to verify correct operation (sporadic flashing), and other functions (charge, low battery, etc.).

Power module: The smart device is battery powered. A standard micro-USB cable is used to charge the device.

Communications module: A Microchip RN4020 transceiver from Microchip implements low-power wireless communications in accordance with the Bluetooth 4.1 standard, using the GATT protocol and features similar to those used in the fall detector to send data in blocks of 20 bytes and receive commands in one-byte blocks.

#### 4.2.2. Elastic Vest

A stretchy vest made of neoprene fabric (95% polyester and 5% elastane) supports the device for respiratory and physical activity monitoring. A first design of the garment was completely closed, as with a T-shirt. A usability test on elderly people highlighted the difficulty of self-placement. The current design is open at the front as a vest and a usability test also carried out on elderly people has shown its adequacy and ease of use, and it doesn’t require the help of another person to put it on. The usability test of the current prototype has also highlighted the comfort of the garment during physical activity exercises. Figure 6 shows a detail of the smart vest placement. To accommodate different chest dimensions, various sizes have been designed (child, XS, M, XXL). The vest adjusts to the contour of the chest thanks to the elasticity of the fabric (double rebound fabric, with elasticity in both the width and length of the fabric). Also, to allow further customization, the vest can be adapted to the user’s chest by adjusting it to various closing positions made of Velcro^®^. Figure 6e shows a detail of the Velcro^®^ adjustment method, with a default position related to size (vertical alignment on the chest in front of the arm) that can be reduced by stretching the closure flap of the vest for a better fit (horizontal alignment under the arm). The usability tests carried out on elderly people of both genders have verified the comfort and fit of the garment to different chest sizes. The vest is washable and has pockets and internal guides that allow for easy donning and, if necessary, removal of the monitoring device, electrodes, and cables. An opening in the position where the device is located allows direct access to the push button and the indicator LED.

#### 4.2.3. The API for Respiratory and Physical Activity Monitoring

The API in the smartwatch provides an abstraction layer between the smartwatch and the smart vest to facilitate the monitoring of respiratory and physical activity. At the programming level, the API provides an instance of the smart vest, which in turn is made up of a set of functional modules that are responsible for managing a series of events associated with the reception of the data sent by the vest. The modules available for the smart vest version for the Android operating system are as follows:

The Bluetooth module notifies of the different states of Bluetooth communication.

The respiratory monitoring module periodically notifies the data associated with respiratory monitoring.

The physical activity module periodically monitors the data associated with physical activity under normal conditions (metabolic expenditure) or during the performance of a particular physical exercise (metabolic expenditure and number of repetitions of the exercise).

Through the FRAIL intervention system, professionals may create exercise plans tailored to specific capabilities and needs of each patient along with assessing their adherence to the plan and performance level. Patients may access the FRAIL gamification system to check their exercise plan and visualize videos and instructions that demonstrate proper performance. In order to execute an exercise, the user selects it (e.g., walking or cycling), and the smartwatch shows the elapsed time since the beginning of the exercise, the number of repetitions performed, instantaneous metabolic expenditure and accumulated metabolic expenditure. Common signals for each exercise have been analysed and incorporated into to the algorithm to identify the activity being performed, which eases counting repetitions (Figure 7). Thus, the device discriminates among activities for total body improvement (e.g., walking or cycling), upper body (e.g., biceps curl, arm flexion, overhead bend) and lower body (e.g., knee extension and flexion, chair push ups, hip flexion, and extension). In resting time between exercises the respiratory rate can be monitored to assess the user’s condition. The adherence to the exercise plan is encouraged by gamification, e.g., by performing exercises three times a week, the player unlocks puzzle-action minigames that provide several hours of gameplay (focused on maintaining cognitive skills). Some of the games have a scoring system, providing some replay ability.

### 4.3. Smartwatch

The smartwatch integrates sensor devices described above in a uniform way with the aim of easing future extensions of the mHealth platform with complementary devices. It also serves as user interface and communications gateway for data storage and remote management of the information. Beyond the activities mentioned above, through the smartwatch the monitored users may access different services:Medication: reminders of pending actions and the history of the medication taken.Agenda: shows pending activities and appointments, as well as a history of past activities.Gamification: reminders of games scheduled. Games are displayed in the gamification system.Breathing: shows real-time data about user’s breathing rate.Heartbeats: displays current heart rate of the user.Alerts: the smartwatch offers the possibility of sending a warning in case of emergency. To alert, the user presses and holds the display for three seconds. A countdown confirms the delivery of the alert.

The bidirectional communication between the devices and the smartwatch is eased by an API deployed in the latter to hide the complexities and heterogeneity of the devices (Figure 8). This element is responsible for receiving/sending data from/to devices, performing the additional processing stage of monitored data, and communicating with the smartwatch.

The API includes a customization module for adjusting algorithms of the processing stage to the characteristics of each subject. Some adjustable parameters are shown in Table 2, including default values. These parameters are stored in a local database in the smartwatch accessible by the API, and those related to device configuration have default values. Beyond that, the proper execution of the API algorithms requires configuring the anthropometric parameters.

## 5. Evaluation

### 5.1. Validation of the Sensors in Laboratory Setting

The fall detection system and the smart vest were experimentally evaluated in a laboratory setting on three male volunteers aged 38 ± 6.2 years (mean value ± standard deviation), 175 ± 4.8 cm of height and 75.7 ± 5.3 kg of weight.

To evaluate the performance of the fall detector, the volunteers carried out a sequence of activities in the laboratory:Fall-free activities: walking, climbing stairs, descending stairs, picking up an object from the ground, sitting in a chair.Fall activities: falling to the floor by first supporting the knees, falling to the floor from a chair, falling to the floor from a bench (simulation of falling from a bed).

The sequence was repeated three times for each of the volunteers. None of the fall-free activities was detected as a fall, and all the fall activities were classified as such, with a sensitivity and specificity of 100%. Although the tests were carried out on young volunteers in the laboratory, the results are promising for the testing with elderly people [26].

To evaluate the precision of the smart vest for respiratory monitoring, the volunteers carried out an experiment in which they voluntarily controlled the inspiration and expiration time at rest, starting with a respiratory rate of six breaths per minute, and progressively increasing the rate according to a pre-established pattern up to 30 breaths per minute. It was not necessary to have a reference standard, since the inspiration and expiration times were defined and synchronized according to a well-defined protocol that allowed a posteriori evaluation of the adequacy of the periods with the respiratory waveforms. In the experiments carried out, the error made in the evaluation of the inspiration and expiration times, as well as the respiratory rate, was negligible since the resulting waveforms were coincident with the respiratory patterns pre-established in the protocol. Although the number of experiments was not very large, the tests served to verify the correct operation of the new smart vest design for the non-intrusive user’s breathing pattern monitoring in the same way of similar monitoring devices [27].

To assess physical activity, the volunteers performed six repetitions of static exercises recommended for elderly people (Figure 9), and one minute of exercise in walking and stationary bike activities. The precision in the evaluation of the number of repetitions was 100% in all static exercises, 98.3% in the count of the number of steps when walking, and 97.5% in the count of the number of strokes in the cycling exercise. This precision is comparable to devices with similar characteristics [28].

### 5.2. Testing of the Devices in a Real Environment

The testing of the devices has been deployed in real environments, reaching a sample of 40 frail and pre-frail elderly persons recruited on a voluntary basis. According to common indicators of frailty assessed by formal caregivers, the inclusion criteria were:over 65 years old,living in an autonomous way but assisted by formal caregivers,risk of falls, andphysical impairments (e.g., slowness or weakness) causing low physical activity.
The exclusion criteria were:
bedridden patients,reduced mobility (wheelchair use), andcognitive impairment.

Participants were randomly assigned to one of the groups (control or intervention).

#### 5.2.1. Key Performance Indicators: Description and Measurement

The USE questionnaire [29] for measuring usability of products as well as services tracks three indicators: usefulness, satisfaction and ease of use, and ease of learning. The survey consists of 30 items, and it has been applied to three elements: smartwatch, fall sensor, and smart vest, resulting in 90 questions answered by each participant. Results are valued between 0 and 7.

The Technostress scale (based on Perceived Stress Scale [30]) evaluates the level of perceived stress during the last month, consisting of 14 items with a response format of a five-point scale (0 = never, 1 = almost never, 2 = occasionally when, 3 = often, 4 = very often). A higher score collected by the direct reckon corresponds to a higher level of perceived stress. This version has been adapted and focused on the specific characteristics of the target group in order to make it more comprehensive and easier to read; only 10 items have been used.

#### 5.2.2. Usability and User Acceptance

The mean scores of the USE questionnaire (4 subscales) after the use of each of the devices are shown graphically in Figure 10. The results show that the smartwatch and the fall detector were valued as complex to learn, which had an effect on the acceptance associated with these technological devices. On the other hand, the smart vest was the best valued in the different usability subscales, especially the degree of satisfaction. This could be due to the fact that this device is less invasive in the daily lives of the elderly adults, whereas the smartwatch requires higher cognitive resources. For all devices, participants showed above average satisfaction, stressing an improvement of safety and sense of control without requiring complex digital skills.

#### 5.2.3. Technostress and USE-Q Correlations

Non-parametric correlations (Kendall’s Tau) were performed in order to explore if the technostress has significant relation with the usability sub-scales in the use of technology.

Results highlight that the technostress variable showed significant correlation with some sub-scales (dimensions) in the USE questionnaire. It negatively correlates with the perceived usefulness (τ = −0.531; *p* = 0.004) and with the satisfaction (τ = −0.479; *p* = 0.01) of the smartwatch. The rest of the USE questionnaire sub-scales applied to each device does not show significant correlation with the technostress variable (values of τ near to 0).

## 6. Discussion

The results obtained highlight the viability of the proposed technology for its application to the monitoring of elderly people. Additionally, these technologies address some of the unsolved issues in the literature.

Numerous technologies have been proposed to detect falls as frail elderly people walk with difficulty and are at high risk of falling [31]. These technologies can be classified into two types: (1) non-portable systems; (2) systems based on sensors and portable devices [32]. Non-portable systems use environmental sensors located in the monitoring area, typically artificial vision systems based on cameras [33] or floor sensors systems (pressure, vibration, capacitive, etc.) [34]. Systems based on cameras and fixed sensors do not protect outside the observation area, requiring a complex and expensive network of cameras and sensors to protect throughout the home, and creating a negative feeling in users when they feel observed/monitored. Portable systems are usually based on motion sensors such as accelerometers and/or gyroscopes. Accelerometry is a very suitable option since it is low-cost, portable and provides information related to movements [31]. Its main disadvantages are related to restrictions on battery life, limited processing capacity and the need for efficient algorithms that detect a fall in a context of use in which motion artifacts can have the same intensity as falls themselves, causing a large number of false positives [26]. Regarding the position of portable sensors, most commercial solutions employ accelerometers on bracelets or pendants that are more sensitive to motion artifacts. In addition, they are placed on visible positions and users are reluctant to use them to avoid stigmatization due to the elder’s weakness. Our detector is located in a position close to the center of gravity (back at the height of the sacrum) and is robust against motion artifacts. In addition, the results of a set of surveys given to elderly people reaffirm its adequacy as it is a non-visible position. On the other hand, devices based on bracelets, pendants or belts are not operative when the person forgets to put them on, is in the shower, or in bed [35]. The proposed device instead allows continuous monitoring (24 h a day), even in the shower or in bed, where most falls occur. To address the high number of false positives that can saturate the system with false alarms, some authors use complex processing algorithms that affect the economic cost and universality of the solution [36]. Other systems assess the immobility of the user after the fall, but attempts to stand up lead to false negatives. Our detector uses a distributed processing algorithm (detection of impacts on the sensor and falls on the smartwatch), capable of distinguishing impact events that can be considered as falls (sitting down abruptly or lying down), minimizing the number of false positives avoiding false negative, reducing energy consumption by not requiring a continuous sending of data and favoring economic sustainability, through the use of a low-cost sensor.

Regarding the smart vest, it is worth highlighting the advantages of the sensorization technology used for respiratory monitoring (capacitive sensorization) [25]. A key aspect is its high sensitivity, and it is capable of detecting variations as small as 20 μm in the spacing of the vest’s electrodes. This translates into a detailed monitoring of respiratory movements, which allows the times of inspiration, expiration and respiratory rate to be measured with greater accuracy and stability, even moreso than the reference methods of capnography or spirometry, since they normally utilize a temporal averaging. The main advantage of the device with respect to reference methods is its comfort, which allows the user to be monitored in a totally transparent way without the user knowing that they are being monitored. The apparatus of the reference methods affects their usability and the feeling of confinement that they can produce would prevent their use in home monitoring conditions. As an alternative to the reference methods, other sensorization technologies have been used, such as accelerometry [37], inductance plethysmography, bioimpedance pneumography [38], measurement of temperature in the nose, photoplethysmography, or ultrasound [39]. However, the precision of these systems is not comparable to that of the reference methods, as in many cases they need to be placed in an uncomfortable or annoying position for the user.

For the evaluation of physical activity, the use of inertial sensors based on accelerometry is common [40]. These devices can provide, in a simple way, numerous metrics related to physical activity such as the number of steps while walking, the time and intensity of physical activity, study of sedentary lifestyle and the estimation of metabolic expenditure [41]. The main disadvantages of these systems are the lack of transparency in the algorithms used by commercial devices, which in many cases are treated as black boxes, and the lack of a gold standard for carrying out activities for the evaluation. The consequences are important differences in the results obtained by different devices in the evaluation of the same physical activity [42]. There is a deficit in the capacity of the devices for their personalization and adaptation to the study of physical activity in people with characteristics different from those used in the validation [43], which can be a strong source in elderly people because they move slowly and the filters used by the algorithms, if they cannot be customized, can remove the components related to movements and activity [40]. In addition, in the context of physical therapy and maintenance of the elderly, the prescribed exercises depend on the conditions and characteristics of each patient, so the needs for customization of algorithms and devices is a pending issue in the literature [44]. On the other hand, in order to promote physical activity in the elderly, systems that allow supervised control of the correct execution of maintenance exercises are of great interest. In this way, patients are motivated to maintain their own health and well-being [45]. For this purpose, a control of the number of repetitions of the exercises is required, commonly applied in pedometers and systems that count the number of steps. However, there are no commercial devices aimed at detecting and recognizing the number of repetitions of other types of exercises such as arm flexion or knee extension, although it is a problem addressed in the field of research [46].

Another approach used in the evaluation of physical activity is based on the processing of images for the capture of movements, as the one used in Kinect. However, these systems have the disadvantage of their low coverage, which extends to the field of vision only [47]. Furthermore, these systems cannot carry out robust activity tracking due to the great variability of postures and movements between different people performing physical exercise [48]. One of the main limitations in the recognition of the activity and the number of repetitions of the exercises is that the recognition models are related to the experiments carried out during training. However, the movements can vary greatly between people in real life. To address this problem, some authors propose complex processing algorithms, usually machine learning techniques, which require a very high number of experiments for their training [44].

The proposed smart vest addresses the problem of monitoring physical activity by solving the issues previously raised by means of algorithms that can be customized to the particular characteristics of the elderly person. Furthermore, and contrary to many of the systems in the literature, the problem of monitoring is approached in a holistic way, not only estimating metabolic expenditure, but also detecting and classifying periods of activity and their intensity, as well as the control of the number of repetitions of the exercises.

## 7. Conclusions

Frailty is a syndrome which may be reduced (even reversed) with the appropriate care plan. This paper presents preliminary results for applying technologies (i.e., FRAIL platform and devices) for empowering pre-frail and frail citizens through a non-intrusive sensing of activity and vital parameters at home. Starting from an elderly-focused design, a set of monitoring devices and a mHealth platform were developed. A technical validation process of the devices and the platform was then performed with young adults prior to involving elderly persons. After that, a set of pre-frail and frail persons was recruited to assess usability and acceptability of the devices. In this assessment, devices were also tested in real (but controlled) environment with real end-users.

This approach’s novelty lies in the combination of different sensors into the same platform for addressing the needs of frail patients. The mHealth platform contributes to remotely monitor and automatically transfer physiological data (heart rate, steps, physical activity, respiratory rates, etc.), and to detect emergencies or risk events such as falls. A dedicated API separates the specific functioning of each sensor from the gateway (here the smartwatch), which is what guarantees the scalability and extensibility of the mHealth platform. Additionally, the two-stage processing allows for the reduction of energy consumption and extending the battery life of the devices. Finally, participants showed positive opinions about the devices, highlighting an improvement of safety and sense of control without requiring complex digital skills. Additionally, it has been found that stress caused by technology may hinder user perception of new solutions, its usefulness, acceptability, and satisfaction. Hence, we can see that the higher the degree of technology phobia, the worse the value of smartwatch usefulness perceived. Similarly, the higher the stress in the interaction with the smartwatch, the worse the satisfaction with it. On the other hand, wearable devices such as the fall sensor and the smart vest do not correlate with the techno-stress variable, resulting in higher usability and acceptance results.

As final remark, some limitations of this work should be noted. First, results show that some components should be modified in order to achieve a higher degree of usability and acceptability before being deployed in real scenarios. Secondly, other devices could be included in the mHealth platform to reinforce the holistic care of frail patients. Finally, the remaining future steps of this work require the performing of a clinical validation in order to quantify the impact of the platform on health outcomes of frail patients.

## Figures and Tables

**Figure 1 ijerph-18-11730-f001:**
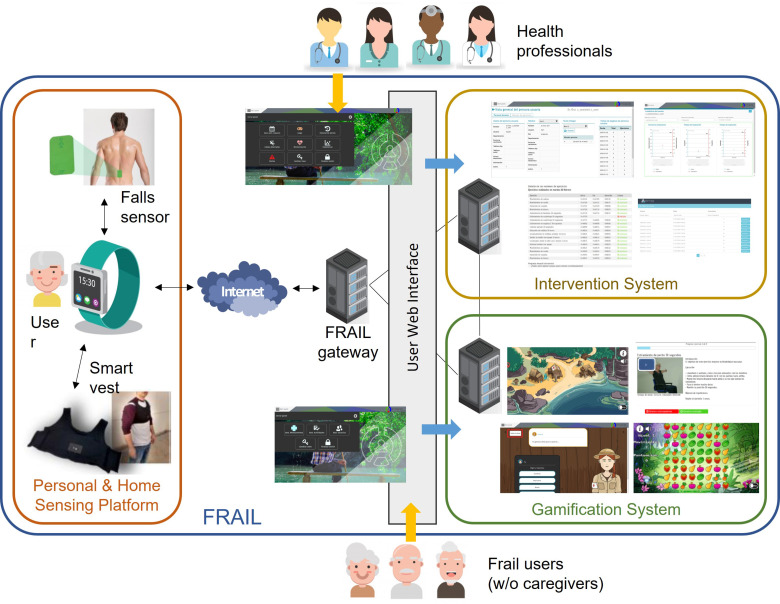
FRAIL architecture and main components.

**Figure 2 ijerph-18-11730-f002:**
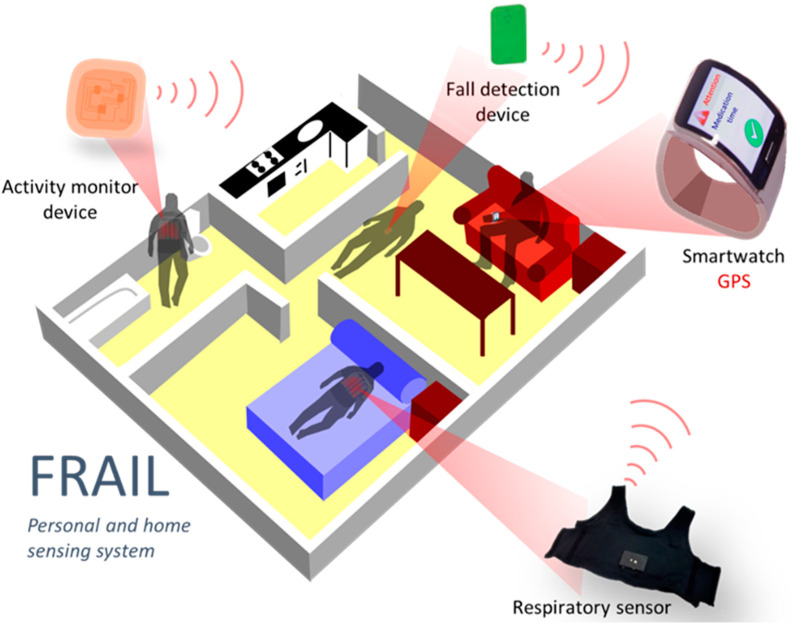
FRAIL Personal and home sensing platform.

**Figure 3 ijerph-18-11730-f003:**
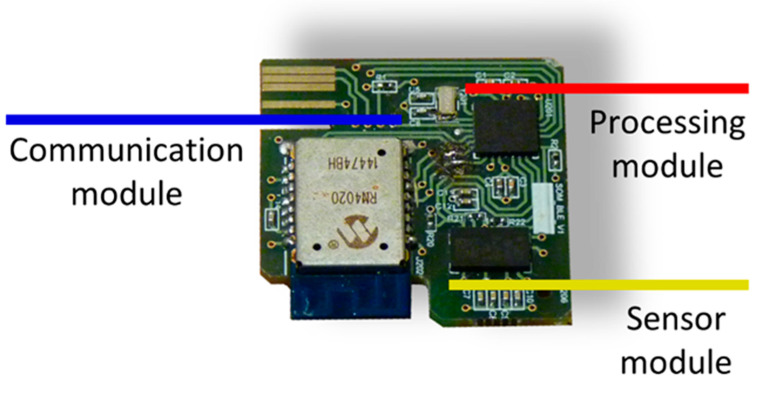
Impact detection device hardware.

**Figure 4 ijerph-18-11730-f004:**
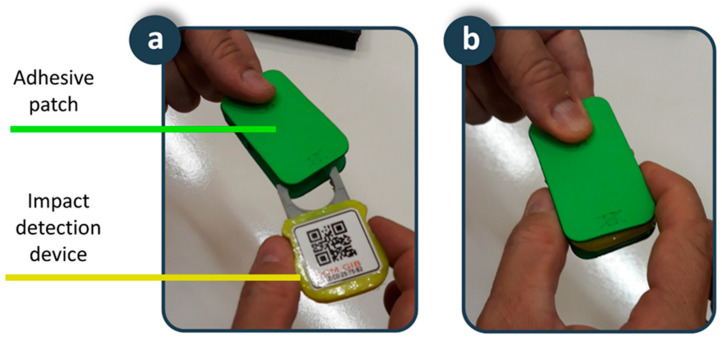
Smart sensor: (**a**) Adhesive patch and impact detection device; (**b**) Detail of the coupling.

**Figure 5 ijerph-18-11730-f005:**
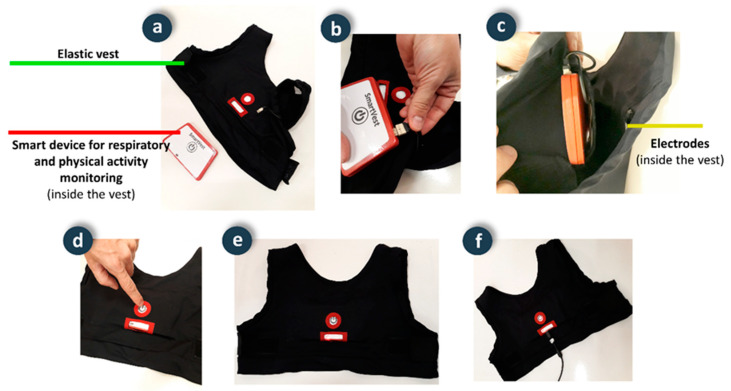
Smart vest: (**a**) Components; (**b**) Device connection; (**c**) Smart vest interior detail; (**d**) Push button power on detail; (**e**) Detail of the indicator led; (**f**) Detail of the charging process.

**Figure 6 ijerph-18-11730-f006:**
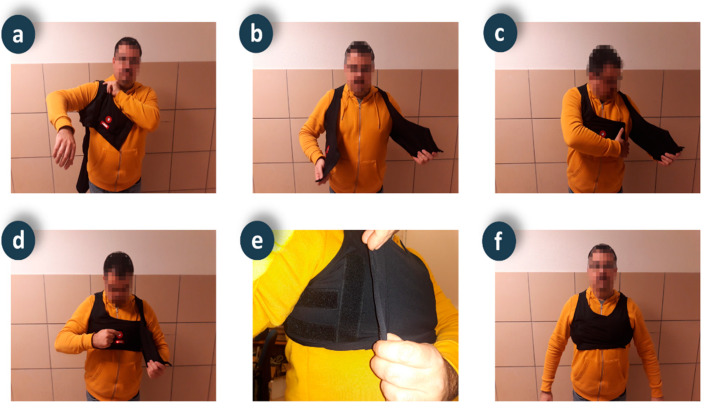
Sequence of the smart vest placement and activation process.

**Figure 7 ijerph-18-11730-f007:**
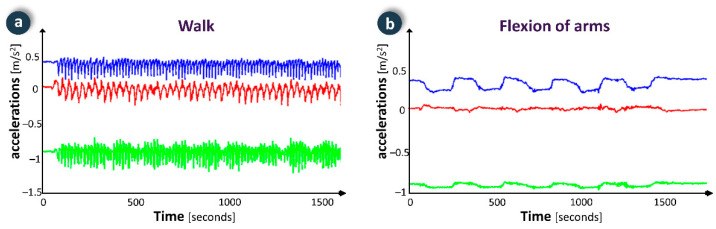
Examples of physical activity monitoring data in two exercises: (**a**) walk; (**b**) flexion of arms.

**Figure 8 ijerph-18-11730-f008:**
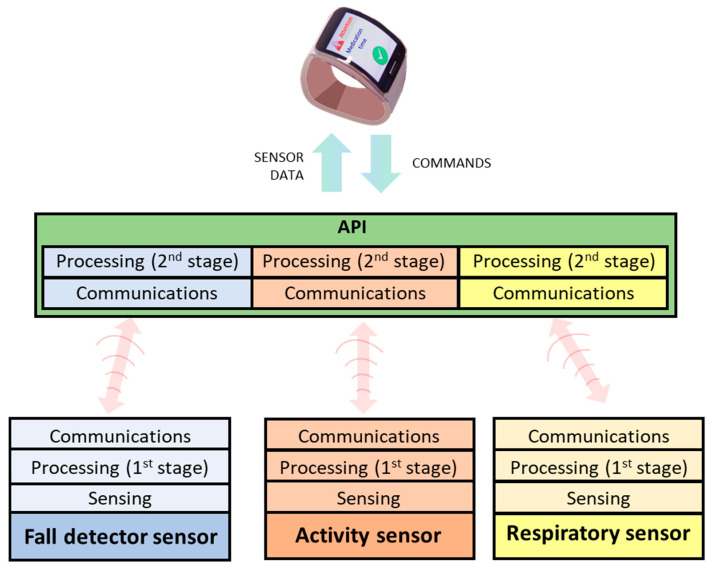
Communication model between sensors and smartwatch through API.

**Figure 9 ijerph-18-11730-f009:**
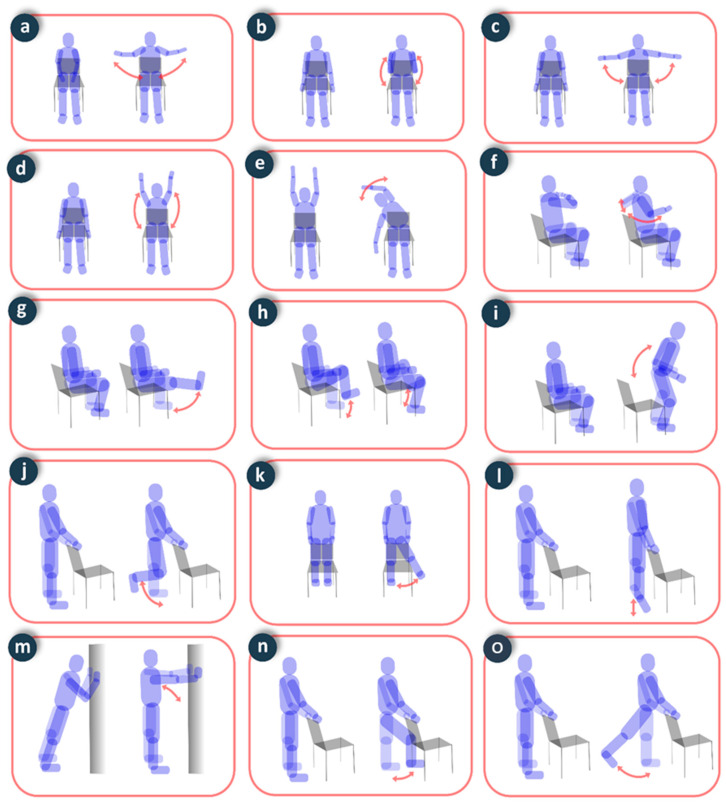
Static exercises performed in the experiments: (**a**) costal expansion; (**b**) biceps curl; (**c**) arm raise; (**d**) arm flexion; (**e**) overhead bend; (**f**) shoulder blade squeeze; (**g**) knee extension; (**h**) marching in place; (**i**) chair push-ups; (**j**) knee flexion; (**k**) side leg raise; (**l**) plantar flexion; (**m**) body inclination; (**n**) hip flexion; (**o**) hip extension.

**Figure 10 ijerph-18-11730-f010:**
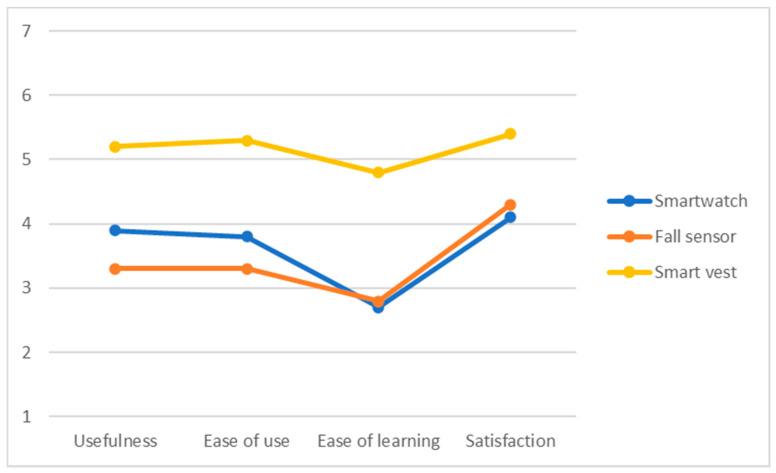
Results of USE questionnaire regarding the smartwatch, fall sensor, and smart vest.

**Table 1 ijerph-18-11730-t001:** User design requirements.

Goal	Need	Design Requirement
Daily life support	Continuous monitoring of vital signs	Unobstructive, portable and ease-to-use sensor devices for heart and breathing rate.
	Risk event support	SOS alert (automated or manual)
	Physical exercise interventions	Gamification platform with personalized exercises
	Maintain adherence to interventions	Feedback and coaching
Avoid falls	Falls detector	Sensor device monitoring physical activity and falls risk
Continuity of care	Remote supervision of health status	Storage of and access to monitoring data
	Notification of event risks	Alert delivery to each stakeholder
	Supervision of physical performance	Automatic performance assessment

**Table 2 ijerph-18-11730-t002:** API personalization parameters.

Category	Parameters	Default Value
Anthropometric	Weight (kg)	N/A
	Birth date	N/A
	Gender	N/A
	Height (cm)	N/A
	Thorax perimeter (cm)	N/A
Fall detection config.	Vertical posture threshold	6.5 m/s^2^
	Fall energy threshold	0.079 m^2^/s^4^
Physical activity config.	Horizontal posture threshold	2.5 m/s^2^
	Movement activity energy threshold	0.1490 m^2^/s^4^
	Resting energy threshold	0.0306 m^2^/s^4^
	Ascending/Descending displacement threshold	0.1/−0.12 m
Respiratory monitoring config.	Intense motion artifact detection threshold	3.7 m/s^2^

## Data Availability

The study did not report any data sets valuable for others.

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
