# Peer review of "A Sensor-Based mHealth Platform for Remote Monitoring and Intervention of Frailty Patients at Home"

_ijerph, 2021, doi:10.3390/ijerph182111730_

Round 1

Reviewer 1 Report

Motion sensors, fall detection sensor studies have been in the healthcare industry for decades and none of them succeeded. Frailty is associated definitely with poor manual dexterity and possible mental incapacity.  Sophisticated HIT gadgets and processes are exceedingly hard to be applied in the elderly though may be successfully tested in the young volunteers. 

There is no good results presentation to convince the readers the project is viable. 

Author Response

  • The authors would like to thank the reviewer for his/her considerated comments that have meant an relevant improvement to this work. In the following, the comments of the review reports are in roman, and the responses in italics.

Motion sensors, fall detection sensor studies have been in the healthcare industry for decades and none of them succeeded. Frailty is associated definitely with poor manual dexterity and possible mental incapacity. Sophisticated HIT gadgets and processes are exceedingly hard to be applied in the elderly though may be successfully tested in the young volunteers. There is no good results presentation to convince the readers the project is viable. 

  • Thank you for the comment. The authors concur with the reviewer’s opinion stating that information technology and devices for the elderly should be carefully designed and tested before their application on real patients. This is our aim in this work. By starting from frail patients’ requirements, we designed non-intrusive and elderly-friendly devices that could mean a high impact on frailty intervention without requiring advanced skills. Beyond elderly-aware design, our results were firstly technically validated by young adults, and then tested with frailty patients to assess their usability and satisfaction. It is worth noticing also that the smart vest received good scores in terms of usability and user acceptance, and satisfaction is above general for all devices. The results of this work motivate to both considering redesign of some components and then resuming with the evaluation of impact in clinical outcomes of frail patients.
  • Taking the reviewer’s comment into consideration we have modified the abstract and conclusions of the paper in order to make clearer the preliminary outcomes of this work, its limitations, and the steps ahead. Furthermore, a new section Discussion has been included highlighting the contributions of this work and its novelty respect to state of the art.   

Reviewer 2 Report

The authors describe the design and testing of a system to monitor frailty among the elderly and to encourage and track physical activity to prevent the condition.

The authors do a great job of describing in detail the design of the system. Clearly, they have put a lot of care in designing devices that are appropriate for the target population.

The pre-testing of the device is also convincing. First, they run tests with younger adults to test sensitivity and specificity. Second they pilot it with some elderly users who they surveyed for their perception of the devices.

My major concern is towards the contribution of the proposed system. The use of sensors on the elderly is not a completely new thing and while the authors mention that their contribution is to not only assess but also prevent, their solution only partially answer that since only one of the device, the vest is really geared towards measuring activities. In that sense, each of the devices in the system should be analyzed in light of the appropriate literature to highlight the novelty and contribution.

Besides, the data collection remains preliminary. The authors only collected perceptual data but it is possible to study the impact of their platform on health outcomes such as frailty.

Overall, this is potentially a good tool that constitutes a helpful development to address frailty but the novelty and actual outcomes need to be better highlighted for the paper to be more conclusive.

Author Response

  • The authors would like to thank the reviewer for his/her considerated comments that have meant an relevant improvement to this work. In the following, the comments of the review reports are in roman, and the responses in italics.

My major concern is towards the contribution of the proposed system. The use of sensors on the elderly is not a completely new thing and while the authors mention that their contribution is to not only assess but also prevent, their solution only partially answer that since only one of the device, the vest is really geared towards measuring activities. In that sense, each of the devices in the system should be analyzed in light of the appropriate literature to highlight the novelty and contribution.

  • In the new version we have added a completely new section Discussion that highlights the contributions of this work and its novelty respect to state of the art. Additionally, we have modified the abstract and conclusions of the paper in order to make clearer the preliminary outcomes of this work, its limitations, and the steps ahead.

Besides, the data collection remains preliminary. The authors only collected perceptual data but it is possible to study the impact of their platform on health outcomes such as frailty.

  • Thank you for the comment. We added a paragraph in abstract and another in the section Conclusion in order to make clear the scope of this work and the steps ahead.

Overall, this is potentially a good tool that constitutes a helpful development to address frailty but the novelty and actual outcomes need to be better highlighted for the paper to be more conclusive.

  • We hope that the new section Discussion satisfies concerns of the reviewer. Thank you.

Reviewer 3 Report

The topic and the proposed platfrom are very interesting and innovativa so I recommend this paper for publication.

However I feel that some relevant changes should be made to the introduction, methods and discussion. Overall this paper in my opinion is very weak on the clinical perspective; possibily, including their reference  geriatrician expert on frailty among the Authors (if they do have one) and having this person review all the issues and references about frailty would greatly help this paper. For example and above all, inclusion criteria for frailty and pre frailty in this population have not even been defined. Also the English writing should be revised,

In detail

LN 33 .....pre-frail persons suffer mild symptoms, and oriented interventions may slow down the decline process. YOU NEED REFERENCES HERE

LN 35-36 Frailty may be considered a state of pre-disability since a situation of incipient functional limitation may lead to develop a new disability, being its importance focused on functionality and not on the diagnosis of a disease. THIS SENTISNCE IN MY OPINION IS A BIT UNACCURATE AND DOES NOT SEEM TO ADD ANYTHING TO THE TEXT. I WOULD SUGGEST TO REMOVE IT

ln 228 4.2.2. Elastic vest. PLEASE SPECIFY HOW THE VEST DEALS WITH LARGE BREASTS

ln 346-7 VERY IMPORTANT: The testing of the devices has been deployed in real environments reaching a sample of 40 frail and pre-frail elderly persons, over 65, living in an autonomous way and without PLEASE ACCURATELY DEFINE INCLUSION CRITERIA FOR FRAILTY AND PRE FRAILTY

LN 388 Frailty is a syndrome which may be reduced (even reversed) with the appropriate 388 care plan. PROVIDE REFERENCES AND ARTICULATE MORE (WHICH INTERVENTIONS HAVE PROVEN EFFECTIVE?

LN 389 This work describes the outcomes of the FRAIL project for empowering pre frail and frail citizens through a non-intrusive sensing of activity and vital parameters at 390 home. IT WOULD BE MORE ACCURATE TO SAY THE PRELIMINARY OUTCOMES, SONCE THE PAPER FOCUSES ON USABILITY, WHILE NOTHING HAS BEEN STUDIED ABOUT THE PLATFORM DIRECT OR INDIRECT EFFECTS ON FRAILTY, WHICH HAS NOT BEEN DEFNIED IN THE FIRST PLACE

Author Response

The authors would like to thank the reviewer for his/her considerated comments that have meant an relevant improvement to this work. In the following, the comments of the review reports are in roman, and the responses in italics.

  • LN 33 .....pre-frail persons suffer mild symptoms, and oriented interventions may slow down the decline process. YOU NEED REFERENCES HERE

Thank you for the comment. We added two references supporting the fact that specific interventions may attenuate or prevent frailty in elders:

  1. Navarrete-Villanueva, D.; Gómez-Cabello, A.; Marín-Puyalto, J.; et al. Frailty and physical fitness in elderly people: a systematic review and meta-analysis. Sports Med. 2021, 51(1), 143-160.
  2. Adja, K.Y.C.; Lenzi, J.; Sezgin, D.; et al. The importance of taking a patient-centered, community-based approach to preventing and managing frailty: A public health perspective. Public Health 2020, 8.
  • LN 35-36 Frailty may be considered a state of pre-disability since a situation of incipient functional limitation may lead to develop a new disability, being its importance focused on functionality and not on the diagnosis of a disease. THIS SENTENCE IN MY OPINION IS A BIT UNACCURATE AND DOES NOT SEEM TO ADD ANYTHING TO THE TEXT. I WOULD SUGGEST TO REMOVE IT

The sentence has been removed. Thank you.

  • ln 228 4.2.2. Elastic vest. PLEASE SPECIFY HOW THE VEST DEALS WITH LARGE BREASTS

Subsection 4.2.2 has been extended to clarify that the vest has been implemented with different sizes and that it includes adjustable Velcro strips and a closure flap to provide a better fit to breast. Thank you.

  • ln 346-7 VERY IMPORTANT: The testing of the devices has been deployed in real environments reaching a sample of 40 frail and pre-frail elderly persons, over 65, living in an autonomous way and without PLEASE ACCURATELY DEFINE INCLUSION CRITERIA FOR FRAILTY AND PRE FRAILTY

A detailed list of inclusion and exclusion criteria for participants in our study has been included in section 5.2. Now this information is shown clearer. Thank you.

  • LN 388 Frailty is a syndrome which may be reduced (even reversed) with the appropriate 388 care plan. PROVIDE REFERENCES AND ARTICULATE MORE (WHICH INTERVENTIONS HAVE PROVEN EFFECTIVE?

As references supporting this sentence have been included in the section Introduction (motivated by a previous comment of the reviewer), we consider it is not necessary to include them once again in Conclusions.

  • LN 389 This work describes the outcomes of the FRAIL project for empowering pre frail and frail citizens through a non-intrusive sensing of activity and vital parameters at 390 home. IT WOULD BE MORE ACCURATE TO SAY THE PRELIMINARY OUTCOMES, SONCE THE PAPER FOCUSES ON USABILITY, WHILE NOTHING HAS BEEN STUDIED ABOUT THE PLATFORM DIRECT OR INDIRECT EFFECTS ON FRAILTY, WHICH HAS NOT BEEN DEFNIED IN THE FIRST PLACE

 Thank you for the comment. We added a paragraph in the abstract and another in the section Conclusion in order to make clearer the scope of this work and the steps ahead.

Reviewer 4 Report

This paper suggests a sensor-based mobile platform integrated into a service-based architecture for remote monitoring and intervention in patients at home. The manuscript presents some flaws that must be addressed by the authors before publication.

1) The abstract should contain additional information such as research methodology and results.
2) A scientific writing review should be considered. The text leaves a lot to be desired.
3) This reviewer misses important articles from recent literature, namely:
"A comprehensive review on smart decision support systems for health care." IEEE Systems Journal 13.3 (2019): 3536-3545. DOI: 10.1109/JSYST.2018.2890121
"Sensor-based characterization of daily walking: a new paradigm in pre-frailty/frailty assessment." BMC geriatrics 20.1 (2020): 1-11. DOI: 10.1186/s12877-020-01572-1
4) A high-level discussion is required concerning the FRAIL project architecture in Section 3.
5) Present in Table 2 the thresholds considered.
6) Improve the Figure 10 presentation.
7) Include the study limitations in the conclusions, as well as suggestions for future work.
8) Some references are incomplete. Correct them!

Author Response

The authors would like to thank the reviewer for his/her considerated comments that have meant an relevant improvement to this work. In the following, the comments of the review reports are in roman, and the responses in italics.

  • The abstract should contain additional information such as research methodology and results.

We added a paragraph in the abstract in order to make clearer the results of this work. Thank you.

  • A scientific writing review should be considered. The text leaves a lot to be desired.

We concur that a scientific literature review is always an interesting starting point of any research, but the scope of this work discourages to do it. Instead, we have supported our contribution with a state of the art (improved in the new version with references [4] and [12]) pointing out the lack of approaches focused on intervention on frailty. Other works have already published literature review in this topic, and it is not our aim to replicate but complement them. 

  • This reviewer misses important articles from recent literature, namely:
    "A comprehensive review on smart decision support systems for health care." IEEE Systems Journal 13.3 (2019): 3536-3545. DOI: 10.1109/JSYST.2018.2890121
    "Sensor-based characterization of daily walking: a new paradigm in pre-frailty/frailty assessment." BMC geriatrics 20.1 (2020): 1-11. DOI: 10.1186/s12877-020-01572-1

Both articles have been exhaustively reviewed and the second one has been included as reference in the manuscript since their relevance on frailty assessment. The first article is interesting, but we consider it is out of the scope of this work. Nevertheless, both references have served as starting point for us to update the state of the art of frailty assessment approaches. Thank you.

  • A high-level discussion is required concerning the FRAIL project architecture in Section 3.

The Section 3 has been extended with more information about the FRAIL architecture as the reviewer recommended. Nevertheless, we purposely aimed not to describe the architecture in too much detail for maintaining the focus of the paper (i.e., exclusively the mHealth platform) and not increasing the length of the manuscript in excess.   

  • Present in Table 2 the thresholds considered.

The applicable thresholds are now available in the revised version of the table. Thank you.

  • Improve the Figure 10 presentation.

Figure 10 has been modified in order to make it clearer. Thank you.

  • Include the study limitations in the conclusions, as well as suggestions for future work.

The section Conclusion has been widely reviewed by including now limitations of this work and future steps. Thank you.

  • Some references are incomplete. Correct them!

All references have been reviewed and completed. Thank you.

Round 2

Reviewer 3 Report

I find the manuscript improved. I am satisfied woth their reply except for one relvant point. I am still not quite satisfied with the inclusione exclusion criteria, that I recommend to be specified more clearly

LN 370-374 living in an autonomous way but assisted by formal caregivers, • risk of falls, and • physical impairments (e.g., low physical activity, slowness, or weakness). The exclusion criteria were: bedridden patients, 374 • reduced mobility (wheelchair use), and 375 • cognitive impairment. HOW WERE RISK OF FALLS AND PHYSICAL/CONGNITIVE IMPAIRMENTS ASSESSED? PLUS LOW PHYSICAL ACTIVITY IS NOT AN IMPAIRMENT. It seems like the only objective criterion was living with a caregiver. tou really need to define bettere frailty and prefraily in your case mix. If you can't you could just say you included elderly persons living with a caregiver, capable of ambulating autonomously and of providing the required feedback. If you did recruit frail and pre frail persons you should state more clearly how they were defined so 

Minor Ln 516 fore involving patients. After that, a set of pre-frail and frail patients was recruited IS IT PERSONS OF PATIENTS?

Reviewer 4 Report

Accept in present form.